# Synthetic Biology Advanced Natural Product Discovery

**DOI:** 10.3390/metabo11110785

**Published:** 2021-11-17

**Authors:** Junyang Wang, Jens Nielsen, Zihe Liu

**Affiliations:** 1Beijing Advanced Innovation Center for Soft Matter Science and Engineering, College of Life Science and Technology, Beijing University of Chemical Technology, Beijing 100029, China; junyang199036@163.com; 2Department of Biology and Biological Engineering, Chalmers University of Technology, 412 96 Gothenburg, Sweden; 3BioInnovation Institute, Ole Maaløes Vej 3, DK2200 Copenhagen, Denmark

**Keywords:** natural products, biosynthetic gene clusters, synthetic biology, genome mining and engineering strategies, design-build-test-learn (DBTL) cycle

## Abstract

A wide variety of bacteria, fungi and plants can produce bioactive secondary metabolites, which are often referred to as natural products. With the rapid development of DNA sequencing technology and bioinformatics, a large number of putative biosynthetic gene clusters have been reported. However, only a limited number of natural products have been discovered, as most biosynthetic gene clusters are not expressed or are expressed at extremely low levels under conventional laboratory conditions. With the rapid development of synthetic biology, advanced genome mining and engineering strategies have been reported and they provide new opportunities for discovery of natural products. This review discusses advances in recent years that can accelerate the design, build, test, and learn (DBTL) cycle of natural product discovery, and prospects trends and key challenges for future research directions.

## 1. Introduction

Natural products (NPs) derived from secondary metabolites of bacteria, fungi and plants have played an important role in traditional drug development. Since the discovery of penicillin and its widespread use as an anti-infective drug, the research and development of NP-derived drugs has opened a rich chapter in the history of human health. NPs and their semi-synthetic derivatives have been playing a crucial role in clinical medicine as antibacterial, antifungal, antiviral, immunosuppressants and enzyme inhibitors [1]. In addition, they are also widely used in agriculture as herbicides, insecticides and fungicides [2]. However, since the 21st century more and more pathogenic bacteria have become drug-resistant, and there is an urgent need to discover NPs with new structures and new biological activities [3].

Traditional NPs discovery strategies, either through chemical synthesis or direct extraction from native hosts, have been successful and discovered many compounds. However, since the 1960s, after a short 10-year golden age, the NP discovery has faced many challenges. A large number of known compounds have been repeatedly discovered. Moreover, traditional methods are inefficient and mostly low-throughput, and could not reach the discovery speed of putative biosynthetic gene clusters (BGCs). Another related challenge is that most BGCs are silent or weakly expressed under laboratory conditions [4], that will increase the discovery cost, mainly on product extraction and detection [5]. Thus, it is urgent to develop new design, build, test strategies that can allow efficient expression and detection of novel NPs.

With the advances of sequencing technology and bioinformatics analysis, a large amount of genome sequence data and putative BGCs have accumulated in public databases. For example, each fungal genome contains 50–90 NP BGCs, which means that these microorganisms have the ability to synthesize 50–90 kinds of NPs [6]. However, the actual number of identified NPs in each genome is far from being exploited. Meanwhile, the gap between the number of predicted BGCs and the identified NPs is still increasing. The rapid development of synthetic biology, including advances in Design-Build-Test-Learn (DBTL) technologies, have greatly enabled mining of novel NPs. The DBTL cycle includes the design of the initial strains or the establishment of a preliminary model system to achieve the determined engineering goals, the construction of the strains, the testing of their outcomes and the understanding of which engineering strategy is effective and why, as well as the incorporation of the learned knowledge into the decision of the subsequent DBTL cycle (Figure 1). Based on the iterative application of the DBTL cycle, academic and industrial biofoundries have been developed to boost the next wave of NP discovery [7]. Here, we discuss recent developments of synthetic biology methods in the design, build, test and learning steps of NP discovery.

## 2. The Design Stage for Natural Product Discovery

In the design stage, it is particularly important to identify genes involved in the synthesis of NPs. A number of databases have been established, including virtualized Registry of Standard Biological Parts (http://parts.igem.org/Main_Page) (accessed on 1 July 2021) that provides a platform for storage, exchange and retrieval of “component” information, KEGG, MetaCyc and BRENDA that store information on known metabolic reactions [8,9,10]. However, many genes involved in the synthesis of NPs are still unknown. Recently developed computational biology techniques provide abundant data and advanced tools for identifying target genes [11] (Table 1).

With the rapid development of gene sequencing and engineering technologies such as molecular biology and genetics, new research ideas and approaches have been applied for NP discovery. The continuous reduction of sequencing costs has made it more and more convenient to obtain genome sequence information of various species. Through the analysis of these genomic data, it is possible to discover, screen and identify potential “silent” gene clusters with novel structures. Therefore, mining for novel active NPs based on massive genomic data has become the focus and hotspot of recent research. With the significant increase in the processing speed and accuracy of DNA sequence analysis, analysis tools based on a large amount of genomic analysis data has been established. For example, BLAST and FASTA can be used to infer the function of unknown genes [12]. In addition to blast based on sequence homology, Hidden Markov model (HMM) analysis of the protein family (Protein family, Pfam) has been widely used [13]. Moreover, the statistical model antiSMASH, which specializes in analyzing and predicting the products of BGCs, has been reported [14] (Table 1). With the significant improvement of sequence processing technologies, the accuracy of protein function prediction has been improved, and the evaluation of BGCs has become more and more accurate [15]. Therefore, the method of mining secondary metabolic biosynthetic gene clusters from genome sequence data is widely used.

Identification of genes encoding secondary metabolic biosynthetic enzymes is a classic method for mining new NPs. Although structures of secondary metabolites are rich and diverse, their biosynthetic mechanisms are relatively conservative, for example, the similarity of amino acid sequences of core enzymes is high [16]. The scaffold core structure of many NPs is polyketide or peptide, which is controlled by the highly conserved polyketide synthase (PKS) and non-ribosomal peptide synthetase (NRPS), respectively. By searching for the biosynthetic genes that control the structure of the scaffold, the biosynthetic gene clusters of NPs can be identified. With the accumulation of biosynthetic knowledge and the advancement of bioinformatics tools and databases, chemical scaffolds of metabolites synthesized by gene clusters can be predicted, and BGCs with new chemical scaffolds can be studied. For example, siphonazole is a NP isolated from *Herpetosiphon* species with anti-plasmodium properties [17]. To track its biosynthetic pathway, genome mining and imaging mass spectrometry technology were used and based on this it was suggested that it belongs to a hybrid polyketide synthase/non-ribosomal peptide synthetase (PKS/NRPS) pathway [17]. On the other hand, if microorganisms are difficult to separate and cultivate, it is not feasible to identify BGCs with the aforementioned methods. An alternative method is to first predict the scaffold structure through bioinformatics, and then chemically synthesize the compound. This method has been used to discover an N-acylated nonapeptide with antifungal activity [18], proving the comprehensive ability of bioinformatics and chemical synthesis in the study of silent gene clusters. However, this strategy is limited to the study of silent gene clusters whose structure can be accurately predicted.

Another method to discover NPs is to analyze not only the coding genes but also the regulatory genes and resistance genes of the BGCs. With the growing understanding of NP biosynthetic pathways, it has been discovered that BGCs not only contain biochemical enzymes, but also regulatory elements, transporters and resistance genes [16]. Therefore, NP mining methods based on resistance or regulatory genes have been developed. For instance, microorganisms that can produce antibiotics have their own resistance systems, which can effectively protect themselves from the synthetic antibiotics. Resistance mechanisms are diverse, including using efflux pumps and degrading enzymes to remove antibiotics, and modifying endogenous proteins to effectively prevent them from binding to antibiotics [19]. The required resistance gene often co-locates with the synthetic gene that encodes the production of antibiotics, so it can be used as a tag to discover putative antibiotics. Using this strategy, a new herbicide was identified by searching the dihydroxyacid dehydratase gene in the published fungal genomes [16].

The discovery of NPs can also be achieved through genome mining based on systematic evolution. The synthesis of new compounds is a very complex process. A recent study showed that by analyzing the evolutionary characteristics of 10,000 BGCs, high-frequency of insertions, deletions and repetitive events occurred in secondary metabolic processes [20]. Two main research strategies based on systematic evolution can be used in NP discovery: One strategy is to construct an evolutionary tree of strains based on conservative housekeeping genes or core genomes, then analyze the NPs produced and identify potential producing strains; the other strategy is to construct an evolutionary tree of genes, and the evolutionary history of the synthetic genes or gene clusters of the products can be inferred from these gene trees. Compared with the method based on the similarity analysis of single sequence, the analysis of biosynthetic function of the enzymes can be more accurate [12].

The discovery of NPs has been boosted through the development of metagenome sequencing and single-cell sequencing. The metagenome includes all the genetic information of the microbial community of both culturable and unculturable microbes. The gene clusters of secondary metabolites are highly repetitive, and sometimes difficult to be analyzed. Through analysis of metagenomics, and the diversity and distribution of NPs in the living environment, it is possible to discover novel substances and their biosynthetic pathways [21]. Moreover, the number of culturable microbes accounts for less than 5% of the total number of microbes. Because traditional microbial technology cannot obtain sufficient quantities of genome DNAs, it is difficult to obtain a large amount of diverse microbial genetic information through genome sequencing technology. The development of single-cell genome sequencing technology provides the possibility to solve this problem [22]. Different from the mass amount of data and complex analysis of metagenomics, single-cell genomes analysis only focus on the genetic characteristics of objects in the most basic biological unit [23]. Single-cell genomics and metagenomics research therefore could complement each other and work in synergy. Single-cell genome mining can directly and accurately discover the evolutionary characteristics and functions of a single cell, while metagenomic mining focuses on obtaining more genetic information on the environmental and evolutional basis. Gene fragments obtained by metagenomics are helpful for pathway prediction in the process of single-cell genome analysis [24]. With the continuous declining of sequencing costs and the continuous upgrading of sequencing technology, it will become easier to mine genetic information in single cells and complex environments, which is of great significance for revealing more putative BGCs.

In addition to identifying putative synthesis pathways, creating pathways not found in nature is becoming a hot research field to discovery new NPs. Combinatorial biosynthesis is based on the versatility of the enzyme substrate, which produces new “unnatural” NPs through the use of engineered catalytic enzymes or metabolic pathways [25,26]. In the process of assembling NPs, the diversity of monomers largely determines the diversity of their structures. The modular type I polyketide synthase (mPKSs) consists of continuous catalytic modules, and each module has a different catalytic region to complete a cycle of C chain extension. For example, in *Streptomyces cinnamonensis*, the polyether antibiotic monensin is biosynthesized through the action of mPKS. The lipid acyltransferase region (AT region) is the fifth module in the monensin synthesis PKS complex, which can absorb unnatural malonic acid derivatives as monomers to synthesize new monensin precursor derivatives. Based on the computer model of the AT region, Bravo-Rodriguez et al. predicted that the AT active region of the enzyme can absorb a larger propynyl group as a substrate monomer, by adding a synthetic compound propargyl-malonyl-N-acetyl-cysteamine, the propynyl-monensin precursor compound was produced by *Streptomyces cinnamonensis* A495 [27]. In the process of polyketide biosynthesis, each module in class I PKS (mPKSs) catalyzes a specific reaction step, and then passes the mature product to the next module. This property makes it possible to design and synthesize new products by “permutation and combination” of these enzyme complexes. Modification of specific enzymes in mPKSs, together with substitution of the substrates catalyzed to produce new compound structures, has become a routine experimental method for combinatorial biosynthesis. For example, using the acetyltransferase region in the rapamycin synthesis pathway to replace the acetyltransferase region in the erythromycin synthesis system, 6-deoxyerythronolide B (6-DEB) analogs containing a variety of new structures have been obtained [28]. Carbohydrates usually play an important role in the drug–target interaction, and the glycosylation process has a significant impact on the solubility and biological activity of the drug. Therefore, the glycosylation of compounds provides a new direction for new drug discovery. For example, the glycosylated NDP-d-digitoxose synthetic plasmid was expressed in *S. argillaceus* M3W1, and seven new structural analogues of mithramycin were obtained by changing the contour of the sugar molecule or changing its molecular contour and 3-side chain at the same time [29], one of which (demycarosyl-3D-β-ddigitoxosyl-mithramycin SK) shows highly effective anti-tumor activities.

## 3. The Build Stage for Natural Product Discovery

Putative BGCs can be evaluated either through activation in the native producer or expression in a heterologous host. In-situ activation of putative BGCs has been greatly advanced by recently developed multiplexed genome editing tools (Table 2). The new generation of genome editing tools based on the CRISPR-Cas technology has the advantages of high efficiency, fast operation and high fidelity. It is the mainstream technology of multiplexed genome engineering at present, especially using the type II CRISPR-Cas9 or CRISPR-Cas12a (Cpf1) gene editing technology. For example, CRISPR-Cas9 mediated knock-in of the *kasO*p* for activation of silent BGCs has been reported and successfully discovered several novel pathways and NPs in Streptomyces species [30]. Moreover, multiplexed automated genome engineering has been reported with the capacity to simultaneously regulate the expression levels of twenty genes, and generate 4.3 billion combinatorial genomic mutations per day [31]. However, the use of CRISPR-Cas technology in most chassis cells requires the introduction of external sources of Cas protein, which can cause cytotoxicity. In order to solve this problem, researchers have developed genome editing technology based on the microbe’s endogenous CRISPR-Cas system in multiple microorganisms, such as *Sulfolobus islandicus*, *Haloarcula hispanica*, *Clostridium tyrobutyricum*, *Clostridium pasteurianum*, *Lactobacillus crispatus* and *Zymomonas mobilis* [32]. These systems allow fast genome engineering including gene insertion, deletion, regulation and single base editing, with the editing efficiency in some cases reaching 100%, and will not be affected by the toxicity of exogenous Cas protein [33].

For microbes with strict cultivation requirements or lack of genetic manipulation tools, heterologous expression of putative BGCs can be used to discovery NPs. Heterologous hosts have many advantages, such as clean background, fast growth and mature genetic manipulation tools. Given that BGCs often include all genes required for biosynthesis of the target NP, cloning of the entire BGC for heterologous expression is of great interest [54]. However, because putative BGCs may have high G + C contents, high sequence similarities and generally large sizes in many cases reaching over 100 kb, selection of the suitable cloning method is crucial. The selection of cloning method depends on the size and complexity of the BGC, whether refactoring is needed, the target NP and the expression host. For example, Polymerase Chain Reaction (PCR) and Gibson assembly-based cloning and refactoring of a streptophenazine BGCs in *S. coelicolor* M1146 resulted in detection of over 100 streptophenazines [54].

Heterologous cloning for NP discovery mainly includes DNA fragmentation, cloning, expression and analysis. According to the method of DNA fragmentation, heterologous cloning can be divided into random library cloning and direct cloning. The random library cloning method constructs expression libraries on random spliced genomes from mixed populations (such as environmental DNAs) or pure cultures, and screens for novel NPs [55,56]. Both sequenced and unsequenced genomes can be used to construct random libraries. The genome is first broken into gene fragments ranging from 10 to 200 kb by partial restriction endonuclease cleavage or mechanical shearing force, that can well cover the size of NP BGCs. However, random library cloning has high chances of disruptions of BGCs, in order to obtain clones containing enough BGCs, it is necessary to obtain a genome coverage of 10–20 times [57]. This need can be solved by optimizing the cloning process, for example, increasing the efficiencies and capacities of DNA extraction, fragmentation, cloning and transformation, avoiding degradations of large DNA fragments and normalizing the cloning and transformation efficiencies among BGCs with varied sizes. After genome extraction and fragmentation, all gene fragments are assembled and transformed into the cloning host for screening [58]. Many NPs have been discovered through random library cloning based on different library construction strategies (such as fosmid, cosmid, phage artificial chromosome and bacterial artificial chromosome (BAC) [55,56]. The random library cloning method is particularly useful when the genome information and features are insufficient. This method has the advantage to cover the entire genetic material [59]. However, this method must rely on high-throughput screening and analytic platforms.

The direct cloning method relies on genome sequencing and bioinformatics analysis to predict BGCs. Target BGCs will then be captured through in vitro CRISPR based digestion or PCR amplification and cloned into target host strains. This method can directly isolate target gene clusters, bypassing libraries construction and the time-consuming and labor-intensive screening process. If refactoring is needed, open reading frames (ORFs) or cDNAs from target BGCs will be amplified through PCR or reverse PCR, respectively and assembled with promoters and terminators from the host strains [60]. If refactoring is not needed, the target BGCs can be achieved through in vitro CRISPR-based digestion. This “molecular scissor” system can specifically recognize target DNAs through user-defined guide RNAs, and achieve efficient and precise cutting [61]. In the past 10 years, the direct cloning method has made great progress [56,62,63]. However, the direct cloning method relies heavily on the quality of genome sequencing and annotation, and can only analyze few gene clusters at one reaction, which greatly reduces the efficiency of NP discovery. With the advancement of synthetic biology tools, it will be ideal that all putative BGCs in one target genome can be cloned at one reaction.

According to the method of DNA assembly, heterologous cloning for NP discovery can be divided into methods for cloning BGCs with or without the need of refactoring (Table 2). The widely used methods for assembly BGCs without refactoring specialize for cloning small fragment number but large fragment size, including Gibson [37], Cas9-facilitated homologous recombination assembly (CasHRA) method [64] and transformation-associated recombination (TAR) [40,65]. Gibson is widely used because of its simple operation and seamless assembly, and it could realize the assemble of four large fragments with the sizes over 100 kb [37]. Compared with the in vitro Gibson assembly, efficient assembly methods of multiple large DNA fragments based on the principle of homologous recombination in vivo have also been popular in commonly used microbial cell factories, including *Saccharomyces cerevisiae*, *Escherichia coli* and *Bacillus subtilis*. For example, the Cas9-facilitated homologous recombination assembly (CasHRA) method has been successfully used in the assembly of large fragments of *E. coli* with a total length of 1.05 Mb, which includes 449 essential genes and 267 growth-related genes [64]. Similarly, TAR has been used to identify several novel NPs, including orphan cosmomycin [66], thiostreptamide and scleroic acid [63]. The methods for assembly BGCs with the need for refactoring feature in cloning multiple fragments with high efficiency and fidelity, including the Golden Gate assembly [34] and LCR assembly [39]. Golden Gate is based on type IIs restriction endonucleases, and can realize combinatorial assembly of 27 components by constructing modular libraries without leaving “scar” sequences [34]. LCR assembly method can realize 20 DNA fragments assembly in one step by using single-stranded bridging oligos between two adjacent fragments of DNA [39]. Combining CRISPR technology with Golden Gate, Gibson or TAR methods has significantly improved the assembly efficiency of large fragments and the size of DNA fragments (1.55 Mb) [25,44]. It will be of great significance to develop more vectors and methods to clone large eukaryotic gene clusters for heterologous expression [67].

## 4. The Test Stage for Natural Product Discovery

The commonly used method for studying NPs is to determine biologically active compounds from the “crude” extract of the fermentation broth, and then fractionate to further separate them. Since this method is time-consuming and labor-intensive, automated liquid handling systems have been developed for high-throughput screening of library-based pre-fractionated crude extracts. Moreover, metabolomics analysis has been developed for simultaneously analyzing large number of metabolites in biological samples. The isomers present in NP extracts can be analyzed using NMR spectroscopy, high resolution mass spectrometry (HRMS), and the LC-HRMS method [68,69]. The advancement of analytical instruments used in NP research, coupled with annotation and calculation methods that can analyze putative NP structures [70], makes the “omics” method more effective.

In order to identify NPs with new structures and activities, it is very important to determine their molecular weights and molecular formulas, and compare them with databases with detailed classification information. However, there are still challenges in data mining and the use of web-based tools to identify metabolites with novel structures [71]. These metadata are sometimes difficult to query from literatures and databases. In this regard, a molecular network platform called the Global Natural Products Society (GNPS) has been developed, and becomes the significant supplement to the toolbox of NP discovery [72]. This network has a large amount of MS/MS data and can visualize the gene cluster of corresponding molecules. Based on these methods, a large number of theoretical NP spectral databases have been created and applied to deduplication [73]. Similarly, METLIN is the platform containing a high-resolution MS/MS database to search for metabolites by analyzing similar structural features derived from reported compounds [74]. Metabolome data together with biological activity analysis can accelerate the identification of biologically active NPs in the extract [75]. Chemometric methods, such as multivariate data analysis, can be used to correlate signals detected in NMR and MS spectra to track active metabolites in complex mixtures. However, current platforms still have limitations, for example, the applicability of certain categories of NPs maybe better than other categories, and there may be ambiguous structure predictions and assignments over certain candidates. Efforts to solve these problems are underway [76], including covering the molecular network of large-scale NP extraction libraries with classification information to improve the credibility of annotation, and the development of comprehensive LC-MS/MS databases to support the NP analysis [77]. Acharya’s team has used this method to characterize the NP-mediated interaction between different species [78]. In general, molecular networks are mainly used to strengthen the deduplication process to better determine the separation priority of unknown compounds and the elucidation of the relationship between NP analogs, and the rigorous structural elucidation of the NPs of interest also need to be improved.

Because the production level of NP is relatively low in most cases, technologies have been developed to increase the detection specificity and tractability of current platforms. With the advancement of N-Methyl-2-Pyrrolidone (NMP) instruments and probe technology, it is possible to analyze a very small amount (less than 10 ug) of the analyte to determine the structure of NP [79]. Analyzing the response of biologically active compounds at the single cell level can also accelerate the discovery of NP drugs. In order to accurately determine the structure of small molecules, a microcrystal electron diffraction (MicroED) based on cryo-electron microscopy has recently been developed [80]. Biology phenotype chip, microplate high-throughput screening, microfluidics, fluorescence activated droplet sorting system (FADS), Raman light spectrum, Fourier transform infrared spectroscopy or near-infrared spectroscopy and advanced spectral sensor have also been used for strain screening and phenotyping profiling [81,82]. For example, Irish et al. combined flow cytometry technology, single-cell chemical biology and cell barcoding with metabolomics to develop a high-throughput platform for bioactive metabolomics analysis [83]. Moreover, combining metabolomics data with transcriptome or proteomics data can also accelerate the identification of NPs [84].

## 5. The Learning Stage for Natural Product Discovery

The learning process involves data collection and integration, data analysis, result visualization, modeling analysis and omics analysis on gene-RNA-protein-metabolism-phenotype. This part provides important feedback for the next cycle and is a critical part in the DBTL cycle of synthetic biology. A large number of omics data and process detection data has been accumulated rapidly, and dedicated public database has provided great convenience for data sharing and storage (Table 1). Meanwhile, databases also provide automated data download programs or scripts, which greatly facilitates the data collection process (Table 1). So far, the data related to non-model microorganisms is developed far less than that of model microorganisms, while the processing and learning of data related to model microorganisms can provide a valuable reference for the study of non-model microorganisms.

The large amount of collected and integrated data can be analyzed using bioinformatics, artificial intelligence/machine learning, especially deep learning, and mathematical models, such as genome-scale metabolic network models and whole-cell models [85,86]. Machine learning obtains capabilities from empirical data and has been widely used in computational biology, metabolic modeling, gene and protein network analysis to guide the design of microbial cell factories. For example, Alphafold developed by machine learning can be used for the de novo prediction of protein structure [87]. Genome-scale metabolic network models are an effective systems biology learning tool, and this kind of model has also been widely used in recent years to quickly understand metabolism of industrial microorganisms [86,88]. Related databases have been established, such as BioModels, BiGG Models and Kbase [89,90,91]. Related data results can promote the further development of synthetic biology through association, centralized query and visualization [92]. At present, many databases provide web-based visualization results, such as BioCyc, which provides tens of thousands of sequenced microbial genomes and their metabolic pathways. The integration of analysis tool and result display is an excellent example of web page visualization [54]. Whole-cell models have also been developed. For example, the whole-cell model of *Mycoplasma genitalium* [93] and a visualization platform WholeCellViz [94] have been established to dynamically display its simulation process, intuitively understand the internal process, and facilitate learning and feedbacks.

## 6. Conclusions and Future Perspectives

The rapid development of genome sequencing and genome mining have become important technologies for NP discovery and drug development [95]. The biosynthetic pathway of NPs involves delicate catalytic mechanisms, regulatory mechanisms and complex metabolic environments. Using bioinformatics analysis and calculation tools can carry out complex data analysis and design to develop NP biosynthetic pathways [96]. Vast genome sequencing data of non-cultivable and cultivable microorganisms, continuous improvement of omics analysis and machine learning technologies, and tools developments of systems biology and synthetic biology will continuously promote the discovery of new types of NPs with biological activity (Figure 2) [97]. In addition, the combination of artificial intelligence and computer methods has been successfully used to predict synthetic modules and pathways [11].

In the process of excavating new structures and new activities of NPs by heterologous expression of silent gene clusters, the selection of a suitable host is also an important factor. Although many unconventional microorganisms have been successfully used as chassis cells, the available modification strategies and tools still lag far behind conventional chassis cells. The newly developed CRISPR system revolutionizes genome engineering of conventional and unconventional chassis cells, and the development of strain-independent genome editing tools plays a very important role in the mining of NPs (Figure 2). The research strategy of genome mining is constantly developing and becoming diversified. In addition to the classic genome mining strategy based on enzyme function analysis, new research strategies such as mining based on system evolution, resistance gene, regulators, culture-independent single-cell and metagenomic have been reported [16]. Nowadays, the acquisition of big data has become easier and easier, but the research on the biological significance behind these data remains a challenge. Effective analytic tools and algorithms will better guide the development of corresponding experiments.

In the mining of BGCs, metagenomics revealed that uncultured organisms can produce NPs with complex structures and biological activities. Using advanced methods and strategies to analyze the human microbiome (including gut microbes, oral microbes and skin microbes) with great biosynthetic potentials will be an important research direction in the future [98] (Figure 2). In particular, the gut microbiota, which is considered to play an important role in health and disease, will be an emerging field for NP drug discovery [99].

Guided by genome mining, in-depth study of the biosynthetic mechanism, regulation mechanism, and key enzyme reaction mechanism is necessary to accelerate the research of NPs. Through genetic manipulation of metabolic pathways in microorganisms, not only more new structures and highly active compounds can be obtained, biosynthetic pathways of different kinds of compounds can also be discovered. In the field of drug discovery and development, combinatorial biosynthesis based on the biosynthesis and metabolism of NPs, can rationalize the genetic modification and reorganization of the biosynthetic pathway at the molecular level, and establish a library of NP analogs with complex structures [100]. Unmodified NPs usually exhibit suboptimal absorption, metabolism, excretion and toxicity properties, and superior analogues with improved pharmacological properties, such as higher specific activity, lower toxicity and better pharmacokinetics need to be acquired in order to yield valuable new drugs [95]. NP analogues can be accessed through the introduction of chemical modifications [101] (Figure 2). From this, drugs with more clinical application values will be developed.

## Figures and Tables

**Figure 1 metabolites-11-00785-f001:**
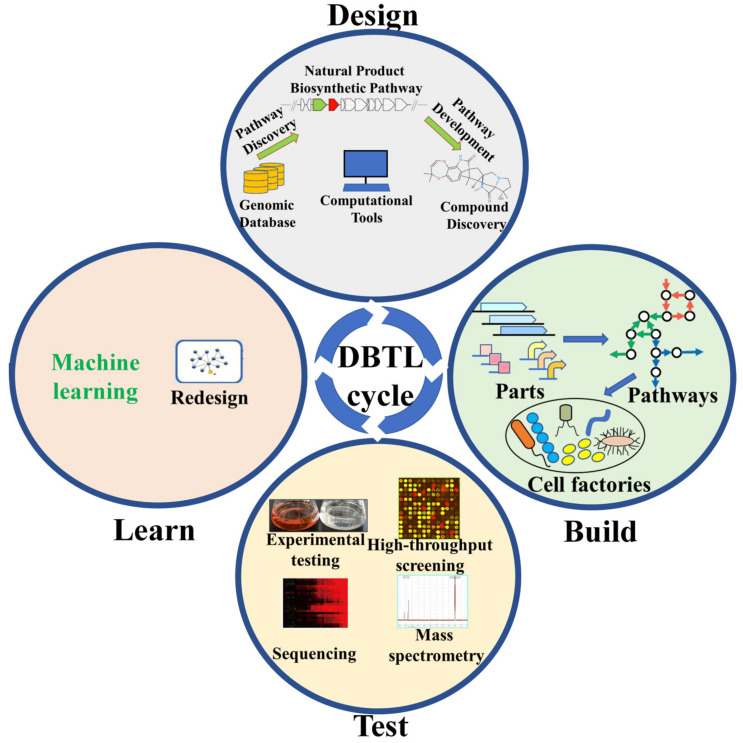
The design–build–test–learn (DBTL) cycle for natural product discovery. Key aspects of each phase of the design–build–test–learn cycle are presented.

**Figure 2 metabolites-11-00785-f002:**
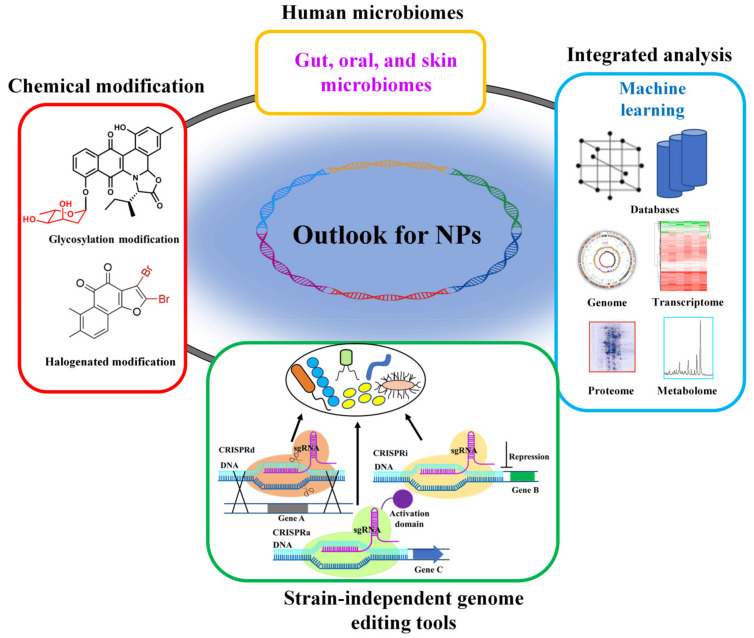
Future perspectives for NPs discovery. In the future, NP repertoire could also be derived from human microbiomes. Existing machine learning methods and databases should be optimized and integrated with the existing omics data, thereby improving the accuracy of the design and accelerating strain development. Strain-independent genome editing tools should also be developed to enable efficient gene editing of nonconventional microbes as well as natural overproducers.

**Table 1 metabolites-11-00785-t001:** Tools for natural product pathway discovery, prediction and analysis.

Name	Description	Web Address
Cluster mining tools		
antiSMASH	Web application to mine and analyze bacterial and fungal genome for secondary metabolite BGCs	https://antismash.secondarymetabolites.org (accessed on 5 July 2021)
Mibig 2.0	A robust community standard for annotation of metadata on BGCs and their molecular products	https://mibig.secondarymetabolites.org/ (accessed on 5 July 2021)
ClusterCAD	A database and web-based toolkit to harness the potential of type I modular polyketide synthases for combinatorial biosynthesis	https://clustercad.jbei.org/ (accessed on 5 July 2021)
PRISM 3	Web for prediction of genetically encoded NRPs and PKs	http://magarveylab.ca/prism/ (accessed on 5 July 2021)
RODEO	Algorithm developed to identify ribosomally synthesized and post-translationally modified peptide BGCs	http://www.ripprodeo.org (accessed on 7 July 2021)
Bagel2	Annotation of putative bacteriocins and antibiotics from genomic DNA	http://bagel2.molgenrug.nl/ (accessed on 7 July 2021)
CLUSEAN	Identification of domains and prediction of specificities for PKS and NRPS genes	https://bitbucket.org/tilmweber/clusean (accessed on 7 July 2021)
SBSPKS	Structural modeling of PKS modules and identification of key residues in the interfaces between modular PKS subunits	http://www.nii.ac.in/sbspks.html (accessed on 10 July 2021)
SMURF	Annotation of PKS, NRPS, NRPS-PKS hybrid, indole alkoloid and terpene BGCs from fungal genomic DNA	http://jcvi.org/smurf/index.Php (accessed on 10 July 2021)
2metDB	A tool offers the possibility to identify PKS and NRPS BGCs	http://secmetdb.sourceforge.net/ (accessed on 10 July 2021)
ClusterFinder	A tool to detect putative BGCs in genomic and metagenomic data	https://github.com/petercim/ClusterFinder (accessed on 10 July 2021)
eSNaPD	A tool to survey BGCs diversity in metagenomic DNA sequences	http://esnapd2.rockefeller.edu/ (accessed on 10 July 2021)
EvoMining	Web for phylogenomics to identify BGCs	http://148.247.230.39/newevomining/new/evomining_web/index.html (accessed on 13 July 2021)
MIDDAS-M	A tool that uses genome and transcriptome data to identify BGCs in fungal genomes	http://133.242.13.217/MIDDAS-M/ (accessed on 13 July 2021)
MIPS-CG	Web application to identify BGCs with genome data	http://www.fung-metb.net/ (accessed on 13 July 2021)
IMG-ABC	Database of experimentally verified and predicted BGCs across 40,000 isolated microbial genomes	https://img.jgi.doe.gov/abc/ (accessed on 13 July 2021)
NaPDoS	Web for offering analysis of PKS/NRPS	http://napdos.ucsd.edu/ (accessed on 15 July 2021)
**PKS/NRPS analytic tools**		
NP.searcher	Web application to identify PKS and NRPS BGCs	http://dna.sherman.lsi.umich.edu (accessed on 15 July 2021)
ClustScan	Web accessible database for PKS/NRPS BGCs	http://csdb.bioserv.pbf.hr/csdb/ClustScanWeb.html (accessed on 19 July 2021)
GNP	Web application to identify BGCs (mainly PKS/NRPS)	http://magarveylab.ca/gnp/ (accessed on 19 July 2021)
NRPS-PKS	Web application to identify PKS BGCs	http://www.nii.res.in/nrps-pks.html (accessed on 19 July 2021)
**Specificity predictors for NRPS or PKS**		
NRPS/PKS substrate predictor	Web for predicting A-domain or AT-domain	http://www.cmbi.ru.nl/NRPS-PKS-substrate-predictor/ (accessed on 21 July 2021)
LSI-based A-domain function predictor	Web for predicting A-domain	http://bioserv7.bioinfo.pbf.hr/LSIpredictor/AdomainPrediction.jsp (accessed on 21 July 2021)
NRPSsp	Web for predicting A-domain	http://www.nrpssp.com/ (accessed on 21 July 2021)
ASMPKS	Web for identification of PKS genes from genomic DNA	http://gate.smallsoft.co.kr:8008/pks/ (accessed on 21 July 2021)
PKS/NRPS Web Server/Predictive Blast Server	Web for predicting A-domain specificities	http://nrps.igs.umaryland.edu/nrps/ (accessed on 21 July 2021)
**Compounds databases**		
ChEBI	A database and ontology of chemical compounds focusing on small molecules	https://www.ebi.ac.uk/chebi/ (accessed on 21 July 2021)
ChEMBL	A database providing information on bioactive molecules with drug-like properties	https://www.ebi.ac.uk/chembl/ (accessed on 21 July 2021)
ChemSpider	A database providing information on structures and properties of over 35 million structures	http://www.chemspider.com/ (accessed on 21 July 2021)
KNApSAcK database	A database on compound information of more than 50,000 natural products of plants and microorganisms	http://kanaya.aist-nara.ac.jp/KNApSAcK/ (accessed on 21 July 2021)
PubChem	A database contains synthetic compounds as well as natural products	http://pubchem.ncbi.nlm.nih.gov/ (accessed on 21 July 2021)
**Metabolomics tools**		
GNPS	Web for analyzing mass spectrometry (MS)/MS data	http://gnps.ucsd.edu/ (accessed on 21 July 2021)
GNP/iSNAP	Web application to automatically identify metabolites in MS/MS data	http://magarveylab.ca/gnp/ (accessed on 21 July 2021)
NRPquest	Web for correlating NRP data with gene clusters	http://cyclo.ucsd.edu (accessed on 21 July 2021)
Pep2Path	Web for correlating peptide sequence tags with NRP and post-translationally modified peptide BGCs	http://pep2path.sourceforge.net (accessed on 21 July 2021)

**Table 2 metabolites-11-00785-t002:** Experimental strategies in the construction of chassis cells for natural products.

Category	DNA Assembly	Description	Reference
**Restriction enzyme-based (in vitro)**	Golden Gate assembly	A method that can assemble multiple DNA fragments using type IIs restriction enzymes	[34]
	Start-Stop assembly	A method that can assemble 60 DNA fragments by type IIs restriction enzymes	[35]
**Homology-based (in vitro)**	One-step sequence- and ligation-independent cloning (SLIC)	A method based on 3′-to-5′ exonuclease activity of T4 DNA polymerase	[36]
	Gibson assembly	A method by T5 exonuclease, Taq DNA ligase and Pfu DNA polymerase	[37]
	T5 exonuclease DNA assembly (TEDA)	A method that requires only T5 exonuclease for assembling multiple DNA fragments	[38]
	Ligase cycling reaction (LCR) assembly	A method that can assemble 20 DNA fragments in one step by introducing single-stranded bridging oligos between two neighboring DNA fragments	[39]
**Homology-based (in vivo)**	Transformation-associated recombination (TAR)	A method depending on the highly efficient homologous recombination system of *S. cerevisiae*	[40]
	Linear-linear homologous recombination (LLHR)	Suitable for cloning small- and midBGCs but require highly specialized capturing vectors and multi-rounds selection.	[41]
	Exonuclease combined with RecET recombination (ExoCET)	A method using short recombination homologous arms.	[42]
**CRISPR (in vivo)**	Cas9-assisted targeting of chromosome segments (CATCH)	A method that can capture 100-kb DNA genomic sequences basing on Cas9 and Gibson assembly	[43]
	Programmable genome engineering	A method that can rearrange 1.55-Mb genome sequences by combining Cas9 and lambda-red recombination	[44]
	Genome editing tools		
**Gene regulation**	Clustered regularly interspaced short palindromic repeats interference (CRISPRi)	The transcription of a gene is repressed by guide RNA and inactive Cas protein	[45]
	CRISPR-AID	A trifunctional system that can simultaneously achieve gene deletion, transcriptional activation and repression	[46]
**Gene deletion**	Multiplex automated genome engineering (MAGE)	Simultaneous editing of multiple genes by prototype devices that automate the MAGE technology	[31]
	Transcription activator-like effector nucleases (TALENs)	Simultaneous editing of multiple genes using TALENs with high portability	[47]
	Zinc finger nucleases (ZFNs)	Allows targeted genome editing but requires re-engineering for every new target site	[48]
	gRNA-tRNA array for CRISPR-Cas9 (GTR-CRISPR)	Simultaneous disruption of 8 genes with high efficiency	[49]
**Gene integration**	Recombinase-assisted genome engineering (RAGE)	Multiplexed integration of large-size DNA constructs	[50]
	Delta integration CRISPR-Cas (Di-CRISPR)	High-efficiency, multicopy, markerless integrations of large biochemical pathways	[51]
**Single-nucleotide conversion**	GTR 2.0	Multiplexed single-nucleotide conversions	[52]
	CRISPR-Cas9- and homology-directed-repair-assisted genome-scale engineering method (CHAnGE)	Rapidly output tens of thousands of specific genetic variants in yeast	[53]

This table is adapted from tables published previously [11].

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
