# Peer review of "Synthetic Biology Advanced Natural Product Discovery"

_metabolites, 2021, doi:10.3390/metabo11110785_

Round 1

Reviewer 1 Report

The authors substantially improved the manuscripts. It is now in condition to be published.

Reviewer 2 Report

The authors have addressed most of the comments satisfactorily. In order to address the comments authors have modified the title of the manuscript as well as significantly improved their tables, figures and the structure of the review. There are a handful that need to be addressed.

Here are my minor comments:

  1. Lines 525 - 527. Add the relevant reference for the Irish et al. work authors refer to in the sentence "For example, Irish et al combined flow cytometry technology, single-cell chemical biology and cell barcoding with metabolomics to develop a high-throughput platform for bioactive metabolomics analysis. "
  2. Line 552 - 553, Page 11. "Results can be feedback to the design stage 552 to further promote the development of synthetic biology through visualization." What kind of results? Results from metabolic modeling or something else? Promote development of synthetic biology only through visualization or also "centralized query"? 
    I would recommend bringing back their previous phrase "promote the development of synthetic biology through association, centralized query and visualization". 
  3. While addressing my previous comment (Point 9) on Page 12, authors mention that they have changed a sentence to "There may be many mutations in the designed biological elements such as enzyme, reporter gene, promoter or ribosome binding site (RBS), designed logical circuits and designed modular metabolic pathways.". I was not able to find the sentence. Has the sentence been completely removed from the main text of the review? 

Author Response

This manuscript is a resubmission of an earlier submission. The following is a list of the peer review reports and author responses from that submission.

Round 1

Reviewer 1 Report

I honestly can't remember the last time I read a manuscript so well done, reasoned and organized that it made me not want to question any of the text. 

The review clearly covers the most modern tools for the research of new drugs using natural products.
My only suggestion is that the images in the manuscript be revised. The ones in the revised manuscript are of poor quality.

Congratulations to the authors. 

Reviewer 2 Report

Comments

This review focus on synthetic biology approaches to accelerate natural product discovery. Although this review is of great interest for the synthetic biology community, there are several points that need to be improved or changed.

Comment 1: Figure 1 quality must be improved as it is a little bit blurred. DBTL should be added to the caption.

Comment 2: Table 1 – PKS, NRPs, RiPPS, MS should be defined. The letter size should be decreased and the information correspondent to each tool must be well defined. The description section should have more information. The table should maybe be separated in sections taking into account the main goal of the selected tools.

Comment 3 – line 195: Remove the word compound.

Comment 4 – Table 2 – The letter size should be decreased and the information correspondent to each tool must be well defined. Define SLIC, TEDA, LCR, TAR, CATCH, MAGE, TALLEN; CRISPR

Comment 5 – Tables- Authors specify that “This table is adapted from tables published previously [17].” However, in some cases the text is exactly the same. Table 2 is currently a reproduction of ref 17. To be accepted the tables need to be substantially different and should contain new and improved information.  The common tools of Table 1 and ref 17 also present exactly the same text.

Comment 6 – Line 355 – Define RBS

Comment 7 – Lines 355-357/Lines 361-363/Lines 363-365/lines 394-399 – Sentences are not ok.

Comment 8 – Line 406 – organism should be in italic.

Comment 9 – line 439 – replace natural products for NPs.

Comment 10 -The English should be significantly improved

Reviewer 3 Report

The manuscript by Wang et al is a well-structured and comprehensive systematic review discussing the recent advances that can accelerate the DBTL cycle for natural product (NP) discovery and challenges and future steps for NP discovery research. The figures and tables are quite descriptive and have covered considerable amount of computational as well as experimental tools available for NP discovery research although there are few concerns and edits that need to be addressed.

Following are my comments:

  1. In introduction, lines 57 – 62, authors mention the latest developments in synthetic biology technologies and emergence of biofoundries. The citations (references 7 to 13) here are not quite recent and I would strongly recommend replacing them with latest literature references. For example: for Biofoundries the latest reference might be Hillson et al, 2019, doi: 10.1038/s41467-019-10079-2.
  2. In Section 2 that describes the design stage for NP discovery, in line 71, what is the reference/citation/link for Registry of Standard Biological Parts? In Lines 75 – 77, authors mention recently develop omics methods and computational biology techniques provide abundant data and advanced tools for identifying target genes. In Table 1 authors describe all the recent computational biology tools and techniques but none of the omics methods are mentioned. What are these omics methods? Recommend rewording the sentence or add relevant information to the table.
  3. I would recommend adding Mibig 2.0 database, org/10.1093/nar/gkz882 and ClusterCAD, doi: 10.1093/nar/gkx893 to Table 1 as they are also equally important tools for Biosynthetic Gene Clusters (BGCs) and Polyketide Synthase (PKS) designing.
  4. In Section 4, test stage, Lines 350 – 352. Reference for recently developed MicroED seems to be missing.
  5. In Section 5, authors describe the learn stage developments. They mention “Genome-scale metabolic network models (GSM), as an effective systems biology learning tool, has also been widely used in recent years to quickly understand metabolism of industrial microorganisms in vivo to find targets for modification…” In addition to reference 86, authors might want to refer to Banerjee et al, doi: 1038/s41467-020-19171-4. This is a recent implementation of GSM driven multiplex CRISPRi strategy for heterologous (high) production of an NRPS in a non-model microorganism.
  6. Reference 37 and 84 seem to be out of place? Reference 37 does not present multiplexed automated genome engineering and reference 84 does not involve barcoding or a polyketide. Might want to remove them or replace with relevant references.

Minor Comments:

  • Title – ….Nature product discovery or …..Natural product discovery?
  • Page 3 line 75 - “computational biotechnologies” should be “computational biology techniques”
  • Page 12, Line 355 – 357 – not clear what authors mean by “…modular metabolic pathways exist a lot of mutants after rational or irrational designing”. Might want to reword it
  • Grammar and spell check
  • Instead of “Irish research team” authors might want to use “first author et al”
  • Line 395 – It should be “systems biology” instead of “system biology”
  • Line 398 – reword “. and related databases are gradually established”
  • Microbiome spelling in figure 2
  • Check reference style/formatting for references 37, 76, 79, 80 and 84